# Developing a national birth cohort for child health research using a hospital admissions database in England: The impact of changes to data collection practices

Ania Zylbersztejn[1,2]*, Ruth Gilbert[1,2,3], Pia Hardelid[1,2]

1 Population, Policy and Practice Research and Teaching Department, UCL Great Ormond Street Institute of Child Health, London, United Kingdom, 2 NIHR Children and Families Policy Research Unit, UCL Great Ormond Street Institute of Child Health, London, United Kingdom, 3 Health Data Research UK London, UCL, London, United Kingdom

* ania.zylbersztejn@ucl.ac.uk

## Abstract

### Background

National birth cohorts derived from administrative health databases constitute unique resources for child health research due to whole country coverage, ongoing follow-up and linkage to other data sources. In England, a national birth cohort can be developed using Hospital Episode Statistics (HES), an administrative database covering details of all publicly funded hospital activity, including 97% of births, with longitudinal follow-up via linkage to hospital and mortality records. We present methods for developing a national birth cohort using HES and assess the impact of changes to data collection over time on coverage and completeness of linked follow-up records for children.

### Methods

We developed a national cohort of singleton live births in 1998–2015, with information on key risk factors at birth (birth weight, gestational age, maternal age, ethnicity, area-level deprivation). We identified three changes to data collection, which could affect linkage of births to follow-up records: (1) the introduction of the *"NHS Numbers for Babies (NN4B)"*, an on-line system which enabled maternity staff to request a unique healthcare patient identifier (NHS number) immediately at birth rather than at civil registration, in Q4 2002; (2) the introduction of additional data quality checks at civil registration in Q3 2009; and (3) correcting a postcode extraction error for births by the data provider in Q2 2013. We evaluated the impact of these changes on trends in two outcomes in infancy: hospital readmissions after birth (using interrupted time series analyses) and mortality rates (compared to published national statistics).

### Results

The cohort covered 10,653,998 babies, accounting for 96% of singleton live births in England in 1998–2015. Overall, 2,077,929 infants (19.5%) had at least one hospital

**Data Availability Statement:** Authors do not have permission to share patient-level Hospital Episode Statistics (HES) data. Qualified researchers can request access to the data from the NHS Digital

Data Access Advisory Group
(enquiries@nhsdigital.nhs.uk).

**Funding:** AZ was funded by a PhD studentship funded from awards to the Farr Institute of Health Informatics Research, London, from the Medical Research Council, Arthritis Research UK, British Heart Foundation, Cancer Research UK, Chief Scientist Office, Economic and Social Research Council, Engineering and Physical Sciences Research Council, National Institute for Health Research, National Institute for Social Care and Health Research, and Wellcome Trust (grant no MR/K006584/1). This research benefits from and contributes to the National Institute for Health Research (NIHR) Children and Families Policy Research Unit, but was not commissioned by the NIHR Policy Research Programme. The views expressed are those of the author(s) and not necessarily those of the NIHR or the Department of Health and Social Care. Research at UCL Great Ormond Street Institute of Child Health is supported by the NIHR Great Ormond Street Hospital Biomedical Research Centre. RG receives funding from Health Data Research UK. The funders had no role in study design, data collection and analysis, decision to publish, or preparation of the manuscript.

**Competing interests:** The authors have declared that no competing interests exist.

readmission after birth. Readmission rates declined by 0.2% percentage points per annual quarter in Q1 1998 to Q3 2002, shifted up by 6.1% percentage points (compared to the expected value based on the trend before Q4 2002) to 17.7% in Q4 2002 when NN4B was introduced, and increased by 0.1% percentage points per annual quarter thereafter. Infant mortality rates were under-reported by 16% for births in 1998–2002 and similar to published national mortality statistics for births in 2003–2015. The trends in infant readmission were not affected by changes to data collection practices in Q3 2009 and Q2 2013, but the proportion of unlinked mortality records in HES and in ONS further declined after 2009.

## Discussion

HES can be used to develop a national birth cohort for child health research with follow-up via linkage to hospital and mortality records for children born from 2003 onwards. Re-linking births before 2003 to their follow-up records would maximise potential benefits of this rich resource, enabling studies of outcomes in adolescents with over 20 years of follow-up.

## Introduction

Administrative health and vital statistics data, including hospital admission and birth and death registration data, are extremely valuable research resources for population health. Whole-country coverage minimises selection bias and loss to follow-up [1], and enables developing "natural experiments" to assess the impact of policy changes and public health interventions on population health [2, 3]. Large sample size enables studying rare outcomes, such as child mortality or congenital anomalies [4, 5]. Further, secondary use of routinely collected data reduces study costs and time compared to de novo data collection. In several countries national birth cohorts for child health research are commonly derived from birth registers (with information about key risk factors at birth, covering all children born in a given jurisdiction), with follow-up from linked death registration and hospital admission records. Such cohorts are commonly used for child health research for example in the Nordic countries [6], Australia [7, 8], Canada [9, 10].

In the UK, national birth cohorts based on birth registration datasets, with follow-up from linked hospital admission and mortality records are routinely used for research in Scotland [11, 12] and Wales [13]. In England, a whole-country birth cohort with rich information about the baby, mother and family could be developed by linking multiple data sources with information about birth and delivery (from Office for National Statistics [ONS] Birth Registration data, National Health Service [NHS] birth notification data, and Hospital Episode Statistics [HES], an administrative hospital dataset), with follow-up via linkage to hospital admission and mortality records. Feasibility of such linkage has been demonstrated by researchers at City University of London [14–16]. These data, however, are only available for births between 2005 and 2014, linkage is not updated routinely, and access is not straightforward [15].

Instead, researchers are increasingly using HES to develop national birth cohorts in England [5, 10, 17] HES covers details of all patient care funded by the National Health Service (NHS) in England, including all births that occur in NHS hospitals (in 2016, 97.4% of all deliveries in England occurred in NHS hospitals) [18, 19]. For each birth admission, HES contains information about the health of the baby at birth and offers the possibility of longitudinal follow-up through routine linkage to consecutive hospital contacts (including admissions) and

mortality records. With on-going data collection and hospital records linkable to an individual reaching back to April 1997, HES has the potential to be an invaluable source of data for child health research, provided there is consistent, high quality of linkage to longitudinal follow-up. However, as with any administrative dataset, changes to data collection methods are outside the control of researchers yet can have profound impacts on the quantity and quality of data collected.

This study aims to evaluate how changes to data collection for HES and vital statistics systems over time have affected the quality of linkage between birth admissions and subsequent HES and mortality records. First, we describe methods for developing a birth cohort using HES. We then demonstrate how changes to collection of patient identifiers used for linkage of birth admissions to subsequent HES and mortality records have affected estimates of readmission and mortality rates. Finally, we suggest how linkage error can be addressed retrospectively to create hospital record trajectories in England for children and adolescents from 1997 onwards with accurate linkage to their birth episodes.

## Methods

### Data sources

**Hospital episode statistics.** HES is an administrative database that covers all hospital activity paid for by the NHS in England (covering an estimated 98–99% of all hospital activity in England) [18]. HES is collated nationally and maintained by NHS Digital. Initially, the database was established to inform management and planning of healthcare services [18]. Since April 2004, data on all admissions are collected for financial purposes [20].

The basic analysis unit in the HES Admitted Patient Care dataset (HES APC) is a consultant episode, defined as the time during which a patient is under the care of one hospital consultant. A hospital admission can consist of multiple consultant episodes if a patient is seen by more than one consultant/healthcare professional [18]. Each HES APC episode includes patient demographic information (e.g.: sex, age, ethnicity, partial post code) and clinical details (codes for diagnoses and procedures) which can be used to derive measures of comorbidities such as congenital anomalies,(4) chronic conditions [21], and long term outcomes (such as emergency admissions [22]). Clinical coders translate information from medical discharge notes into diagnostic codes (using the International Classification of Disease version 10, ICD-10 [23]) and procedure codes (using a UK-specific system, the Office of Population Censuses and Surveys Classification of Interventions and Procedures, OPCS [24]). Although all NHS clinical coders are required to complete national accreditation training to ensure standardisation of recorded information between hospitals, coding sensitivity could vary according to the quality and level of detail covered in discharge notes [18]. Socioeconomic status is measured using the Index of Multiple Deprivation (IMD) score, an area-level measure of deprivation allocated at the Lower Layer Super Output Area level (covering on average 1500 people) according to the patient's postcode. The score combines area-level indicators in seven domains: income, employment, health and disability, education, crime, barriers to housing and services, and living environment [25].

HES APC also includes the 97% of births in England that occur in NHS hospitals (but not home births or births in private hospitals). Each birth in HES APC leads to at least two records: one delivery episode for the mother and a birth episode for each baby. Both the maternal delivery and baby's birth episodes contain the same 19 variables detailing the delivery and labour, called the "baby tail", including information such as gestational age, birth weight, maternal age, mode of delivery etc. Delivery and birth episodes are not routinely linked by NHS Digital [18].

HES has been collected nationally since 1989. Patients' hospital admission records can be linked over time since 1st April 1997 using the HESID–a study-specific pseudonymised patient identifier generated by NHS Digital [26]. The algorithm to generate HESID is based on the NHS number, local hospital patient identifier, date of birth, postcode and sex (see S1 Appendix Table A for details) [27].

**HES-ONS linked mortality data.** Information about causes and date of deaths can be obtained through linkage of HES to Office for National Statistics (ONS) mortality records, which cover all deaths registered in England. For deaths at ≥28 days of life, information about the underlying cause of death, and any additional conditions that contributed to death are recorded. For deaths at age <28 days, the neonatal death certificate is used, which lists main conditions in the baby and the mother, with no single underlying cause of death [28].

NHS Digital links HES episodes to ONS mortality records for deaths registered from the 1st January 1998 onwards using NHS number, date of birth, sex and postcode (see S1 Appendix Table B for details) [29].

NHS Digital also flags all in-hospital deaths recorded in HES (that is, hospital episodes where the discharge method was recorded as 'died'). NHS Digital compares information for in-hospital deaths recorded in HES and ONS mortality records and provides an indicator of agreement between these sources. Deaths identified using HES only (even if there is no link to an ONS mortality record) are included in the HES-ONS mortality data. However, these deaths do not have any recorded causes of death available [29].

## Study participants

We developed a birth cohort of singleton live-born infants, who resided in England and were born between 1st January 1998 and 31st December 2015, based on birth admissions recorded in HES. We identified birth admissions using broad selection criteria based on diagnostic codes and admission details recorded in HES (such as admission type, see S2 Appendix Table A for details). We used only information recorded in baby's birth record, as linkage between maternal delivery and baby's birth admissions is not routinely available. We excluded multiple births due to an increased risk of false matches (i.e. multiple individuals being allocated the same HESID, meaning that it is not possible to distinguish between their hospital or mortality records) among same sex siblings. We also excluded stillbirths and births indicated by unfinished HES episodes, which should not include any clinical details (detailed exclusion criteria are listed in S2 Appendix Table B). We cleaned data on key risk factors recorded at birth admission in HES (birth weight, gestational age, sex, ethnicity, maternal age, IMD score and county of residence), excluding implausible or inconsistent recordings, and finally we removed infants who were not resident in England. Details of data cleaning are described in S2 Appendix, and Stata code for cohort derivation can be found at https://github.com/UCL-CHIG/HES-birth-cohorts.

Children were followed up from birth to their first birthday or death by linkage to hospital admission and mortality records using HESID. All episodes of care in HES APC were linked into hospital admissions, which we defined as a continuous period of time that a child spent under NHS hospital care. Hospital transfers and admissions within 1 day of each other were treated as one inpatient admission [30].

## Outcomes

We focus on assessing trends in two outcomes which could be affected by changes in data collection practices for patient identifiers used for linkage between birth admissions and follow-up records. Firstly, we looked at the hospital readmission rate in the first year of life to assess

impact of changes in data collection on internal linkage between birth and hospital admissions within HES. We defined the hospital readmission rate as the proportion of infants with at least 1 hospital admission in the first year of life after discharge from the birth admission. The birth admission included transfers to neonatal intensive care units or other hospitals within 1 day of birth. Second, we looked at the infant mortality rate to evaluate of the impact of changes in data collection on external linkage to ONS mortality records. Infant deaths were indicated if a linked ONS mortality record was found or if discharge method in hospital record indicated death. We calculated the infant mortality rate as the number of children who died at age 0–364 days per 1000 live births.

## Changes in collection of patient identifiers used for linkage

We identified three changes in data collection which could affect the completeness and accuracy of patient identifiers recorded in HES and influence linkage within HES and with other datasets:

1. **Q4 2002:** implementation of NHS Numbers for Babies (NN4B) service on 29th October 2002. Since October 2002 maternity staff in England request an NHS number for babies in hospital after birth using an on-line system as part of the Statutory Birth Notification process. Prior to this point babies were allocated NHS number by registrars at birth registration, which could take up to 6 weeks [31].

2. **Q3 2009:** Introduction of Registration ONline system (RON) on 1st July 2009 RON is a web-based online registration system for births and deaths used by registrars in local authority registry offices. The introduction of RON enabled additional validation checks at the point of birth and death registration, such as validation of address and postcode [32].

3. **Q2 2013:** correction of a postcode extraction error by NHS Digital on 1st April 2013 An extraction error resulted in birth admissions missing postcode prior to the 2013/14 financial year. As a result, all consultant episodes specified as births (using *epitype* variable in HES) were missing a postcode and all variables derived from postcode (including IMD score) [33].

## Statistical analysis

**Cohort coverage.**   We estimated the coverage of the HES birth cohort by comparing the number of births per year in the cohort to national birth statistics for England published by the ONS [34, 35]. We derived the proportion of births per calendar year with valid and complete information on key risk factors recorded at birth: birth weight, gestational age, sex, ethnicity, maternal age, and quintile of IMD score as a proportion of all births in the HES birth cohort.

**Impact on internal linkage between births and hospital admission records.**   We used interrupted time series analysis (ITSA) to assess the impact of changes in data collection on internal linkage of birth records to hospital admission records [36]. We fitted a linear regression model with the proportion of babies with at least one hospital readmission after birth per annual quarter of birth as the outcome. We adjusted for year of birth (as a continuous variable) and indicator of annual quarter of birth. We also included three binary indicators for time periods before and after each of the changes to data collection equal to 0 before and 1 after change (to quantify changes in observed readmission rates in Q4 2002, Q3 2009 and Q2 2013 vs "expected" rates for these time points based on the trends before each change). Lastly, we also adjusted for an effect modification term between year of birth and the period indicator

before vs after each change to data collection (to assess changes in the trends in readmission rates in Q1 1998-Q3 2002, Q4 2002-Q2 2009, Q3 2009-Q1 2013 and Q2 2013 onwards).

We hypothesised that the quality of identifiers, and therefore linkage, would be better for babies who have a longer birth admission as their NHS number and other identifiers can be updated during their hospital stay. Therefore, we repeated the models separately for readmissions in babies with long birth admission ($\geq$7 days) and in babies with short birth admission (<7 days).

**Impact on external linkage between births and infant mortality records.** We compared trends in crude infant mortality rates derived from the HES birth cohort with official infant mortality statistics published for singleton live births in England and Wales by the ONS (95% of births in England and Wales occur in England) by year of birth to evaluate the impact of changes in data collection on linkage between HES and ONS mortality records [37–39]. We tabulated mortality by age at death as neonatal mortality (at 0–27 days) and post-neonatal mortality (at 28–364 days).

Deaths in linked HES-ONS data can be defined through a link to ONS mortality records or via the discharge method recorded in hospital, and all in-hospital deaths should link to ONS mortality records. To further explore the impact of the three changes to data collection on linkage between HES and ONS mortality data, we looked at changes by year of birth in: (1) the number and proportion of deaths in the HES birth cohort that were identified using only discharge method in HES (with no ONS mortality record), and (2) the number and proportion of infant deaths from ONS mortality records which did not link to any HES record.

All deaths with an ONS mortality record should have recorded causes of death. We explored the proportion of these deaths in the HES birth cohort with no recorded causes of death as an indicator of potential data extraction errors. All analyses were done using Stata version 15.

### Ethics statement

We have a data sharing agreement with National Health Service (NHS) Digital to use a de-identified extract of HES data linked to ONS mortality records for research on child health. We did not require ethical approval to use these datasets [40].

## Results

### Cohort coverage and data completeness

We identified 10,653,998 singleton live births in HES APC in calendar years 1998–2015, covering 96.4% of all singleton live births in England (Table 1). Overall, birth weight was complete for 66% of births, gestational age for 63%, maternal age for 62%. Completeness has improved over time, reaching on average 82% for birth weight, 78% for gestational age, and 69% for maternal age for births from 2009 onwards (Fig 1). Baby's sex was over 99% complete throughout the study period. IMD score (based on earliest recording in any hospital admission record in the first year of life) and ethnicity (based on most commonly recorded value) were complete for 53% and 74% of births, respectively. IMD score was complete for only 34% of births in 2008–2012 reflecting the NHS Digital data extraction error that led to postcode and all postcode-derived variables being missing from birth admission records. The error was corrected in April 2013. IMD score was complete in 89% of records in calendar years 2014–15.

### Impact on internal linkage between births and hospital admission records

Overall, 2,077,929 (19.5%) babies had at least one hospital readmission after birth during infancy. The proportion of babies with hospital readmission after birth declined by 0.2%

**Table 1. Coverage of HES birth cohort compared to national statistics published by the ONS for England.**

| Year of birth | HES birth cohort | England (according to ONS)* | coverage |
|---|---|---|---|
| 1998 | 546,804 | 584,928 | 93.5% |
| 1999 | 542,734 | 572,611 | 94.8% |
| 2000 | 526,254 | 556,172 | 94.6% |
| 2001 | 527,602 | 547,292 | 96.4% |
| 2002 | 540,030 | 549,003 | 98.4% |
| 2003 | 554,104 | 572,711 | 96.8% |
| 2004 | 571,530 | 589,248 | 97.0% |
| 2005 | 577,964 | 595,019 | 97.1% |
| 2006 | 592,768 | 616,588 | 96.1% |
| 2007 | 602,801 | 635,561 | 94.8% |
| 2008 | 631,027 | 652,280 | 96.7% |
| 2009 | 630,170 | 649,416 | 97.0% |
| 2010 | 649,431 | 665,746 | 97.5% |
| 2011 | 650,223 | 666,320 | 97.6% |
| 2012 | 653,116 | 672,505 | 97.1% |
| 2013 | 626,619 | 646,941 | 96.9% |
| 2014 | 614,637 | 640,663 | 95.9% |
| 2015 | 616,184 | 643,363 | 95.8% |
| **total** | **10,653,998** | **11,056,368** | **96.4%** |

HES = Hospital Episode Statistics. ONS = Office for National Statistics.

*We estimated the number of singleton live births in England based on the number of total births in England [34, 35] and assuming that the ratio of singleton live births to all live births was the same in England as in England and Wales (97.0% in 1998–2015) [37–39].

percentage points per annual quarter from Q1 1998 until Q3 2002 (Fig 2). In Q4 2002, after the NN4B system was introduced, the hospital readmission rate increased by 6.1% percentage points (compared to the "expected" readmission rate based on the trend before Q4 2002) to a total of 17.7%. Thereafter, the readmission rate increased by 0.1% percentage points per annual quarter for births in Q4 2002 –Q2 2009 and in Q3 2009 –Q1 2013, and by 0.2% per annual quarter from Q2 2013 onwards. There was no statistically significant change in the observed vs expected proportion in Q3 2009 when RON was introduced nor in Q2 2013 when the postcode extraction error for births was corrected by the data provider. Detailed ITSA results are shown in S3 Appendix Table A.

Results separated by length of birth admission were comparable, suggesting that linkage error was not affected by length of stay. The proportion of babies with a long birth admission (≥7 days) who had at least one hospital readmission in infancy after birth declined by 0.2% percentage points per annual quarter between Q1 1998 and Q3 2002. In Q4 2002 when NN4B system was introduced the readmission rate increased by 8.6% percentage points to a total of 29.8% (compared to the "expected" readmission rate based on trend in 1998–2002), and continued to increase by 0.2% percentage points per annual quarter in Q4 2002 –Q2 2009, and by 0.1% thereafter. There was no statistically significant change in the observed vs expected readmission rate in Q3 2009 nor in Q2 2013. For babies with a short birth admission (<7 days), the readmission rate declined by 0.2% of births per annual quarter of birth in Q1 1998 –Q3 2002. In Q4 2002, the observed readmission rate increased by 5.9% percentage points to a total of 16.7% of babies with hospital readmission compared to the "expected" rate and increased thereafter. There was no statistically significant change in the trend in Q3 2009 nor in Q2 2013.

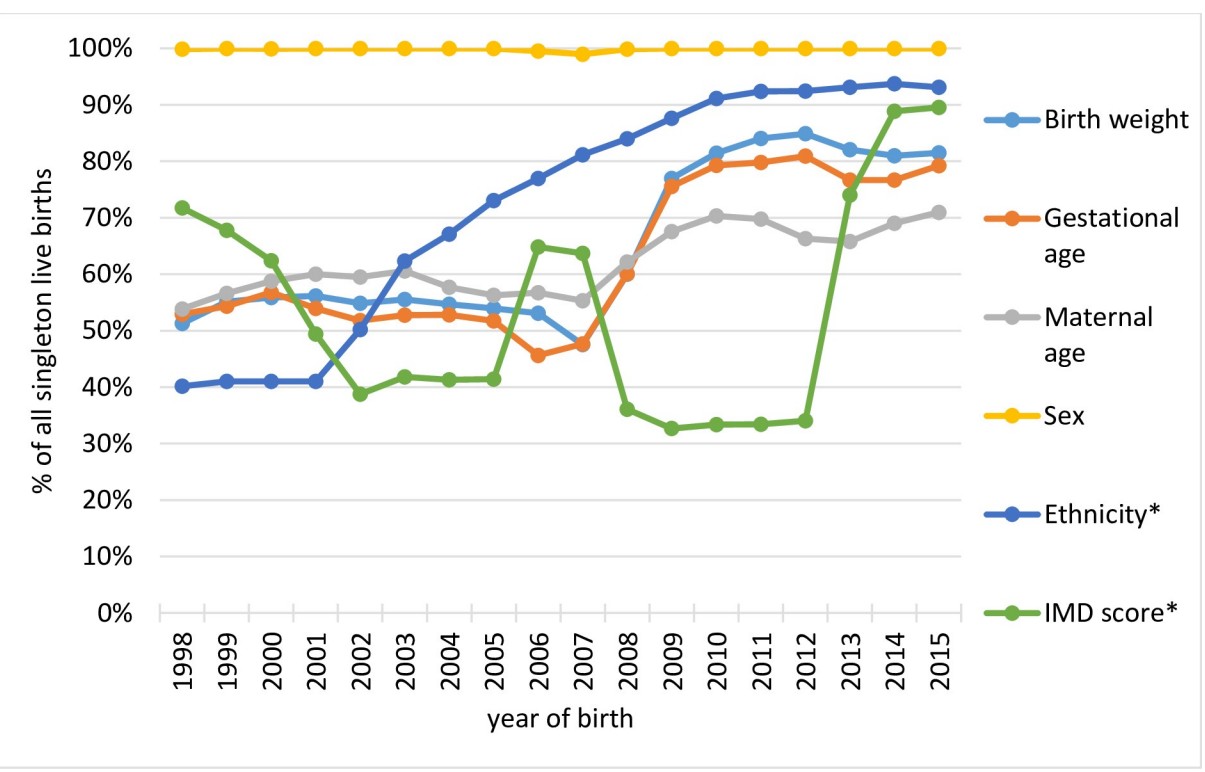

**Fig 1. Completeness of key risk factors recorded in baby's birth records in HES birth cohort.** HES = Hospital Episode Statistics, IMD = Index of Multiple Deprivation. *Note that ethnicity and IMD score were completed using each child's longitudinal hospital admission records (see S2 Appendix for details).

## Impact on external linkage between births and infant mortality records

We identified 42,963 infant deaths between 1998 and 2015 in the HES birth cohort. Infant mortality in the cohort was 4.03 deaths per 1000 live births in 1998–2015, compared to 4.16/1000 live births reported by ONS for England and Wales. Infant mortality rates were underestimated by 16% for births before 2003, and closely matched rates reported by the ONS for births in 2003–2015 (Fig 3). The difference in rates before 2003 was primarily driven by under-reported post-neonatal mortality rates in the HES birth cohort. Given 2,683,424 births in 1998–2002 identified in HES, the difference in mortality rates derived from HES birth cohort and national statistics reported by the ONS translates to 2,082 deaths missing from the HES birth cohort, of which 1,347 were in the post-neonatal period.

Overall, 13% of deaths in the HES birth cohort were identified using only information recorded in hospital admission records, with no link to an ONS mortality record. The proportion of deaths with no linked ONS mortality record decreased over time: before introduction of NN4B system in 2002, 29% of infant deaths had no link to an ONS mortality record. This proportion decreased to 10% between 2003 and 2009 (when RON was introduced) and further to 5% in 2010–2015 (Table 2). There was no change after 2013 when postcode extraction error was corrected by NHS Digital. Similar patterns were observed for the proportion of infant deaths recorded in ONS mortality rates that did not link to any HES record (Table 2).

80% of deaths with no ONS mortality record occurred in the first week of life (*n* = 4,548). Because there is no linked ONS mortality record, these deaths do not have any recorded causes of death in the linked HES-ONS data. We found that a further 1% of all deaths with a linked

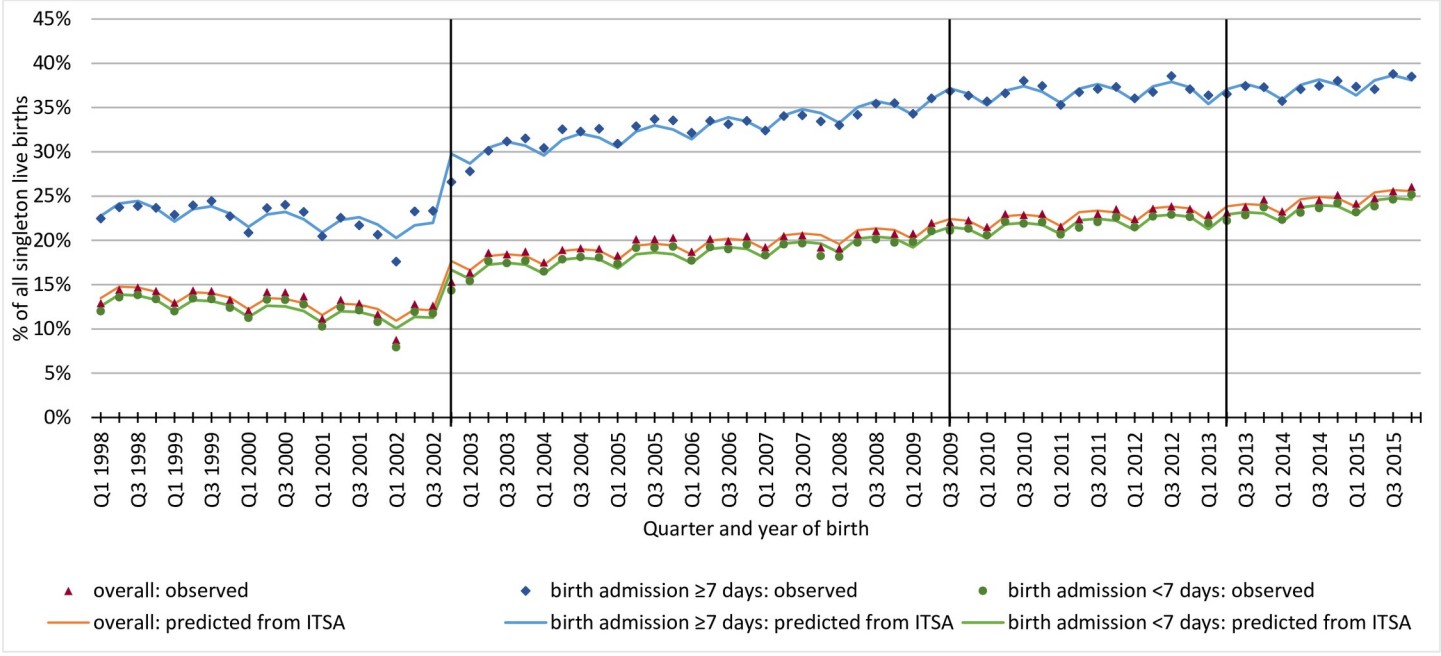

**Fig 2. Trends in the proportion of infants with at least one hospital admission after birth in infancy by quarter and year of birth.** HES = Hospital Episode Statistics, ITSA = Interrupted time series analysis. Vertical lines indicate time points when the collection of identifiers used to generate HESID has changed: 1) Q4 2002: implementation of NHS Numbers for Babies service 2) Q3 2009: Introduction of registration online system 3) Q2 2013: correcting postcode extraction error by NHS Digital.

ONS mortality record had no recorded causes of death. 98% of these deaths were at age 28–30 days, accounting for 75% of all deaths on days 28–30 in 1998–2016.

## Discussion

### Key findings

The implementation of NN4B, which allowed the NHS number to be allocated to babies in hospital shortly after birth led to substantial improvements in ascertainment of subsequent hospital admissions and infant deaths in a birth cohort developed using HES birth records. Unreliable linkage between births and follow-up records before 2003 resulted in a misleading downward trend in hospital readmissions before the introduction of NN4B in October 2002. Infant mortality rates derived from the HES birth cohort were also underestimated for births in 1998–2002. The introduction of RON correlated with a reduction in the proportion of unlinked deaths indicated in HES and ONS mortality records. Fixing the postcode extraction error in 2013 did not impact the quality of linkage, but it helped to improve the completeness of the IMD score for birth records in HES.

### Strengths & limitations

We present validated methods for developing a national birth cohort using HES, which covered 96.4% of births in England. Whole-country coverage enabled us to assess impact of changes in data collection on linkage to follow-up records as the trends in mortality or hospital admission records were not affected by loss to follow-up or selection bias. To ensure we captured all births, we used broad selection criteria for identifying births including diagnostic and procedure codes, and administrative variables recorded in HES. We recommend this approach

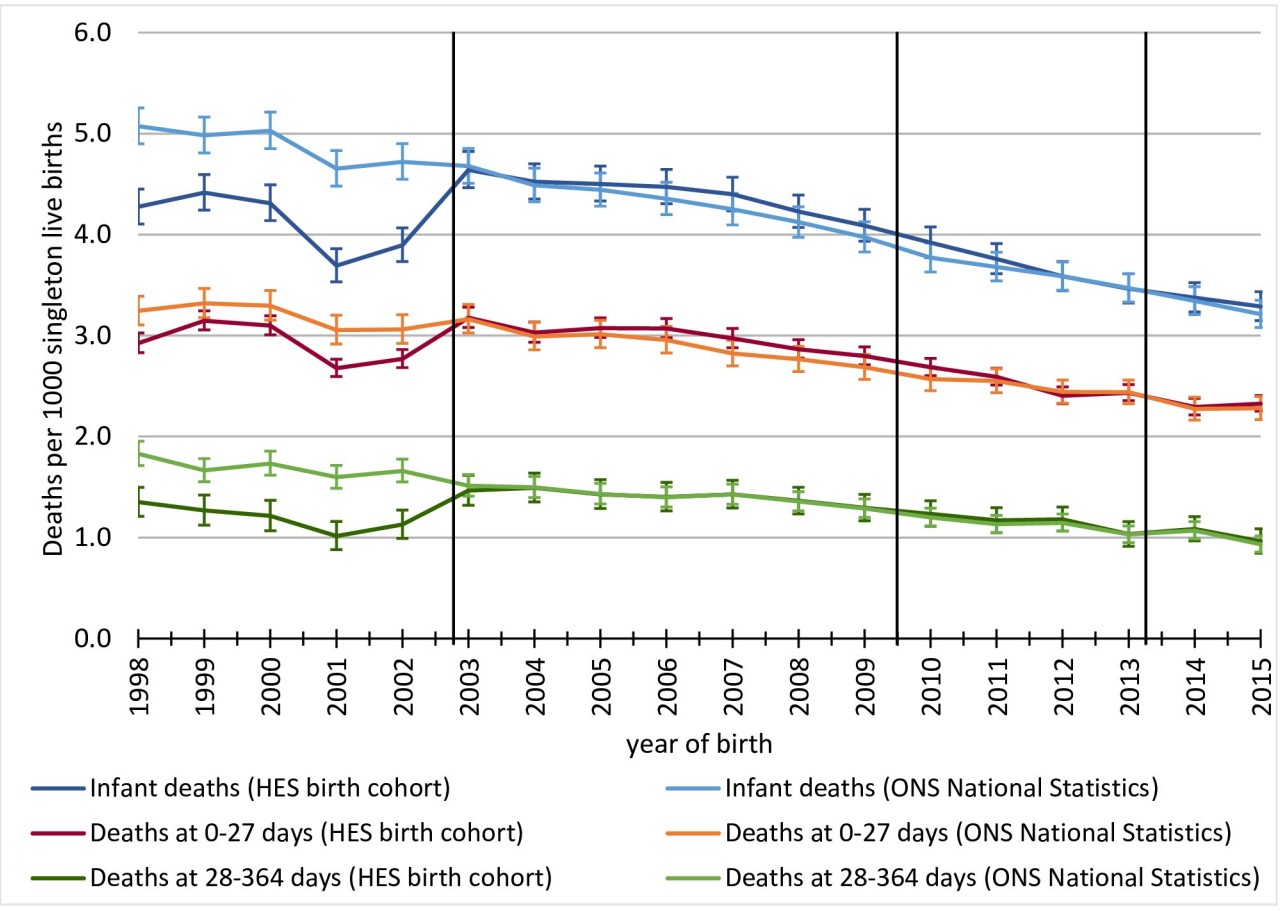

**Fig 3. Comparison of components of infant mortality in HES birth cohort compared to national figures reported by ONS.** HES = Hospital Episode Statistics, ONS = Office for National Statistics. Vertical lines indicate time points when the collection of identifiers used to generate HESID has changed: 1) Q4 2002: implementation of NHS Numbers for Babies service 2) Q3 2009: Introduction of registration online system 3) Q2 2013: correcting postcode extraction error by NHS Digital.

to maximise cohort coverage. For example, previous studies using only admission method to indicate births identified 87% of births [41], while an algorithm based on multiple variables captured 97% of births [42]. Our criteria were consistent with previous studies (with minor differences reflecting differences in study aim) [42, 43]. Our Stata code is openly available on Github, with detailed description for users in S2 Appendix of this paper. Our paper can serve as a helpful primer for researchers interested in using HES for child health research.

Our methods excluded multiple births from the cohort, due to the increased risk of false matches for multiple births, especially for same sex siblings [27], and increased uncertainty about coding of stillbirths among multiple births [42]. Our exclusion criteria (listed in S2 Appendix Table B) could be used to develop a cohort of multiple births. Further work is needed to evaluate the coverage of multiple births in HES, and coding of stillbirths for multiple births in the HES birth cohort. A recent national audit of maternity and perinatal care for women with multiple births and their babies has identified case ascertainment of 89.5% using multiple detailed datasets [44]. Further work is needed to evaluate the quality of linkage to longitudinal hospital admission and mortality records among twins and triplets.

We did not have individual-level information about personal identifiers recorded at each admission. To evaluate the quality of linkage we were therefore limited to looking at trends in

**Table 2. Number and proportion of infant deaths that did not link to ONS mortality record and number and proportion of infant deaths that were indicated using HES data only.**

| Year of birth | HES birth cohort | | ONS Mortality Records | |
|---|---|---|---|---|
| | Total infant deaths* | % without a link to ONS mortality record | Total infant deaths | % without a link to HES |
| 1998 | 2,337 | 33% | 4,357 | 28% |
| 1999 | 2,396 | 24% | 4,584 | 26% |
| 2000 | 2,269 | 27% | 4,358 | 27% |
| 2001 | 1,948 | 33% | 4,320 | 30% |
| 2002 (introduction of NN4B) | 2,104 | 32% | 4,246 | 29% |
| 2003 | 2,571 | 12% | 3,967 | 14% |
| 2004 | 2,585 | 10% | 3,864 | 13% |
| 2005 | 2,602 | 9% | 3,870 | 11% |
| 2006 | 2,651 | 9% | 3,931 | 11% |
| 2007 | 2,651 | 10% | 3,969 | 11% |
| 2008 | 2,668 | 10% | 4,106 | 12% |
| 2009 (introduction of RON) | 2,577 | 8% | 3,985 | 8% |
| 2010 | 2,546 | 6% | 3,834 | 7% |
| 2011 | 2,444 | 6% | 3,656 | 7% |
| 2012 | 2,342 | 4% | 3,509 | 8% |
| 2013 (correcting postcode extraction error for births in HES) | 2,171 | 5% | 3,181 | 8% |
| 2014 | 2,075 | 4% | 2,934 | 8% |
| 2015 | 2,026 | 4% | 2,924 | 7% |

HES = Hospital Episode Statistics, NN4B = NHS Numbers for Babies (NN4B) ONS = Office for National Statistics, RON = Registration ONline system. Rows marked in grey indicate years when data collection practices have changed.

*Infant deaths were indicated if a linked ONS mortality record was found or if discharge method in hospital record indicated death.

two outcomes likely to be affected by changes in data collection for HES and vital statistics. Information on the proportion of babies with missing NHS number, postcode and sex per calendar year could be provided by NHS Digital to researchers when data are provided, allowing the assessment of potential biases in their studies due to linkage errors.

## Interpretation

The introduction of NN4B in October 2002 had the biggest impact on the quality of linkage between birth, hospital and mortality records. Prior to the introduction of NN4B in October 2002, newborns were assigned an NHS number at birth registration, which could occur up to 6 weeks after birth. If a birth episode did not contain an NHS number, it could only be linked to consecutive admissions in HES and to ONS mortality records using postcode, date of birth and sex and no link would be established if, for example, the child changed their address. As a consequence, babies born before October 2002 were more likely to be allocated a new HESID at first hospital admission following birth.

Re-linking birth episodes to longitudinal hospital admissions prior to 2003 would provide a unique resource for birth cohort studies of health outcomes in adolescents with 20 years of follow-up after birth. There are two barriers to re-linking births before 2003 to their consecutive records. First, many births might be missing NHS numbers as discussed above. Secondly, due to a data extraction error by the data provider, birth episodes prior to April 2013 are missing postcode at birth [33]. Re-linking these births to consecutive hospital admissions and death

records would therefore require three steps. First, babies need to be linked to their mothers to obtain information about postcode at birth, as postcode is 99% complete in mothers' delivery records in HES [42]. Second, using the date of birth, sex and complete postcode at delivery, birth episodes would need to be linked to the Personal Demographics Service (PDS), a national database of all patients who interact with the NHS (including all patients registered with a GP, babies who have received an NHS number at birth, as well as patients admitted to hospital via accident and emergency), and its predecessor the National Health Service Central Register (NHSCR) [45]. Finally, the NHS number obtained via linkage between HES birth records and PDS/NHSCR could be used to re-link HES birth episodes before 2003 to HES admissions after birth. However, more support and resources would need to be allocated to NHS Digital to address these data quality issues.

We also identified a need to improve the completeness of risk factors recorded at birth admissions in HES. Key risk factors recorded at birth admission (birth weight, gestational age, maternal age) were recorded in 62–66% of births. It has previously been shown that completeness and accuracy of these baby variables in HES vary between hospitals (as submission of these variables to HES is not mandatory), and several approaches have been developed to derive and validate nationally representative sub-cohorts of births in hospitals with high quality of recorded data [5, 41, 46, 47]. Due to a data extraction error, the IMD score–the only measure of socioeconomic status available in HES–was missing from birth episodes prior to April 2013. Completeness of key risk factors at birth could be improved by linking mother's delivery and baby's birth admission records as maternal delivery records are often more complete than birth records, and have not been affected by the postcode extraction error prior to 2013. Mother and baby records can be linked using deterministic and probabilistic methods (with a linkage rate of 96%) [42]. We have previously shown that copying information from the maternal delivery record to the linked baby birth record result in a high completeness recording of key risk factors including birth weight and gestational age [48].

Complete information on key risk factors at birth could also be obtained through linkage of HES to ONS birth registration and NHS birth notification data. These datasets contain highly complete information on a number of risk factors including birth weight, gestational age, parity, parental country of birth and ethnic group [49]. Feasibility of linkage between these datasets and HES has been demonstrated by researchers at City University of London [14–16]. Since these datasets are now all held by NHS Digital, this valuable linkage could be routinely updated [14]. Linkage of multiple children to the same mother through HES, ONS birth registration, NHS birth notification or all of these datasets, would additionally enable characterisation of siblings and family groups [50, 51].

Since April 2015 maternity and child health services in England are also required to contribute data collected in antenatal clinics (such as smoking status or body mass index at first booking), and details of delivery and birth collected at the maternity ward (such as gestational age, delivery method, and diagnoses of the newborn baby) to the Maternity Services Data Set (MSDS) also maintained by the NHS Digital [52]. This dataset is not yet linked to HES and the quality of recorded data has not been evaluated nationally. Given sufficient data quality, linkage of MSDS and HES would provide a rich resource for perinatal health research in the future.

National birth cohorts from administrative linked datasets provide an invaluable resource for child health research. Many countries (e.g. the Nordic countries, Scotland or Australia) have a long tradition of using such data for research, with data collection reaching back to the '80s. We demonstrate that in England, HES can be used to develop a birth cohort, following children born from 2003 into their adolescent years. A birth cohort for children born in 1998 onwards would be possible to develop if errors in linkage of birth episodes were corrected. Birth admissions cover key characteristics of mothers and babies, such as birth weight,

gestational age, sex, maternal age and area-level deprivation, comparable with information recorded in birth registers in the Nordic countries, Scotland, Canada and Australia, although improvements to data completeness are needed. Diagnoses are coded using ICD-10 classification, which is also used in hospital records in Europe, New Zealand, Australia and Canada [18], enabling international comparisons of child health outcomes (such as mortality, congenital anomalies, chronic conditions or respiratory tract infections) [4, 5, 8, 10, 53].

## Conclusion

We identified a significant improvement in linkage within HES records and to ONS mortality records with the introduction of a programme to allocate NHS numbers at birth in 2002. HES provides a unique resource for future child health studies, with ongoing data collection, and historical data going back to 1998, allowing over 20 years of follow up for the oldest children in the cohort. To fully benefit from this rich resource for child health research, improvements in the quality of recorded data are needed. Linkage of babies' birth record to mothers' delivery records, ONS birth notification and NHS birth registration data can be used to enhance the completeness of key risk factors at birth. Birth admissions prior to 2003 need to be re-linked to consecutive admissions and death records. Such re-linkage would be invaluable for birth cohort studies of health outcomes in adolescents and adults where long follow-up times are needed.

## Supporting information

**S1 Appendix. HES linkage algorithms.**
(DOCX)

**S2 Appendix. Developing a cohort of singleton live births in HES.**
(DOCX)

**S3 Appendix. Additional results.**
(DOCX)

## Acknowledgments

This work uses data provided by patients and collected by the NHS as part of their care and support.

## Author Contributions

**Conceptualization:** Ania Zylbersztejn, Ruth Gilbert, Pia Hardelid.

**Data curation:** Ania Zylbersztejn.

**Formal analysis:** Ania Zylbersztejn.

**Funding acquisition:** Ruth Gilbert, Pia Hardelid.

**Investigation:** Ania Zylbersztejn, Ruth Gilbert, Pia Hardelid.

**Methodology:** Ania Zylbersztejn, Ruth Gilbert, Pia Hardelid.

**Resources:** Ania Zylbersztejn, Ruth Gilbert, Pia Hardelid.

**Software:** Ania Zylbersztejn, Pia Hardelid.

**Supervision:** Ruth Gilbert, Pia Hardelid.

**Validation:** Ania Zylbersztejn.

**Visualization:** Ania Zylbersztejn.

**Writing – original draft:** Ania Zylbersztejn.

**Writing – review & editing:** Ania Zylbersztejn, Ruth Gilbert, Pia Hardelid.

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
