## [Decision Letter · Decision Letter 0]

21 Jul 2020

PONE-D-20-14030

Developing a national birth cohort for child health research using a hospital admissions database in England: the impact of changes to data collection practices

PLOS ONE

Dear Dr. Zylbersztejn,

Thank you for submitting your manuscript to PLOS ONE. After careful consideration, we feel that it has merit but does not fully meet PLOS ONE’s publication criteria as it currently stands.

Major objections have been raised during the review process about the methods and attainable aims of the described process and database: Although some of them may be out of reach given the setting of the study, we invite you to consider submitting a revised version of the manuscript that addresses the remarks made by the reviewers. 

Would you choose to do so, please address all the points made by the reviwers in the below report. 

We look forward to receiving your revised manuscript.

Kind regards,

Umberto Simeoni

Academic Editor

PLOS ONE

Journal Requirements:

2.We note that you have indicated that data from this study are available upon request. PLOS only allows data to be available upon request if there are legal or ethical restrictions on sharing data publicly. For information on unacceptable data access restrictions, please see http://journals.plos.org/plosone/s/data-availability#loc-unacceptable-data-access-restrictions.

Reviewers' comments:

Reviewer's Responses to Questions

**Comments to the Author**

1. Is the manuscript technically sound, and do the data support the conclusions?

Reviewer #1: Partly

Reviewer #2: Yes

2. Has the statistical analysis been performed appropriately and rigorously? 

Reviewer #1: Yes

Reviewer #2: Yes

3. Have the authors made all data underlying the findings in their manuscript fully available?

Reviewer #1: No

Reviewer #2: No

4. Is the manuscript presented in an intelligible fashion and written in standard English?

Reviewer #1: Yes

Reviewer #2: Yes

5. Review Comments to the Author

Reviewer #1: This is a manuscript describing hospital episode statistics (HES) data on births in the UK and assessing whether this data can be used for following up children using routine data. While this paper provides interesting descriptive information about England’s HES statistics, the research question (s) are difficult to discern. It reads, at times, more like a report describing the constitution of a dataset, than a scientific study. The authors need to be more straightforward about the central premise of their study, clarify the research aims and place the results in a broader international context.

The introduction and title of the MS place emphasis on the methods for identifying a “national birth cohort” – the authors begin by citing the failure of the 2015 effort to establish the Lifestudy and then refer to examples from other countries where register data are used for research. From the onset, the authors need to clarify what they mean by “national birth cohort”. There is a large difference between being able to link hospital data and studying some longer-term (mainly hospital-related) outcomes and creating a national birth cohort (as initiated in many countries, see https://lifecycle-project.eu/). When the authors claim in their discussion that they have shown the feasibility of using HES to create a national birth cohort, this is not convincing if they mean this cohort to be an alternative for cohort studies such as Lifestudy or if comparing with the Nordic registers. Many other issues remain outstanding such as whether data can be linked to other sources, issues of consent, etc…

The research questions underlying the study aims are not clear. It is stated: “In this study, we present methods for developing a national birth cohort using HES and provide Stata code for cohort derivation. We demonstrate how long-term health outcomes of children in the cohort (such as hospital admission or death) are affected by changes in the quality of recorded identifiers used for linkage of birth admissions to consecutive hospital admissions within HES and to other datasets. We suggest how linkage error can be addressed retrospectively to create hospital record trajectories for children and adolescents from 1997 onwards with accurate linkage to their birth episodes.

Concerning the first aim, presenting methods and giving code is transparent and helpful for other researchers, but does not constitute a research question. So, although instructions to identify births are provided, there is no test to show how this definition improves on others. There is no discussion about whether these methods are similar to those used in other datasets (for instance: Kuklina EV, Whiteman MK, Hillis SD, et al. An enhanced method for identifying obstetric deliveries: implications for estimating maternal morbidity. Matern Child Health J. Jul 2008;12(4):469-477). Most studies using hospital discharge data use algorithms to identify births.

Second, the section on how changes to the quality of recorded identifiers improved linkage and quality of indicators relying on linkage isn’t particularly surprising and therefore, while it is reassuring to find that better linkage seems to improve the accuracy of some indicators (leading to higher admission rates, for instance), it’s not clear how this adds to overall knowledge about how hospital episode statistics can be used for research on children. Furthermore, there is no gold standard - in the case of hospital admissions – and there may still be substantial errors.

In terms of the conceptualisation of the study, the database constituted by the authors only allows a partial evaluation of the capabilities of the HES data as mothers are not linked with their babies, even though the authors state that this is possible and would improve the quality of data. The authors need to justify why they developed this study without linking these data.

Given the large amounts of missing data, it is not clear why the authors did not describe the characteristics of births with and without missing data – is this related to the hospital? The region?

The reader also wonders whether there has there been any validation of these data with medical records to assess the validity of the data?

Many countries use hospital discharge data for research; putting the HES within this broader international context would be useful. How do these results compare to those in other countries?

Abstract –“Numbers for Babies (NN4B) system for allocation of unique National Health Service (NHS) number at birth in Q4 2002” is not interpretable as a stand alone sentence.

While multiples pose many problems for linkages and use of administrative and register data, the solution of eliminating them is not optimal.

Reviewer #2: This study described methods used to create a national birth cohort using HES and aimed to evaluate the quality of linkage between births and follow-up records and its impact on two health outcomes in children. Overall, I think this is a well written and informative study.

To help further improve the manuscript it would be good if the authors could address the following points:

- It’s great to see that the authors have provided a link to the Stata code for derivation of the cohort. Can they just check that this is complete and consistent with all the steps they describe in the paper e.g. from a quick scan of the Stata code I could not see any code for excluding non-English residents. Similar, it would be really helpful if they can make sure that all the data cleaning steps in the Stata code are described in the appendix of the paper e.g. from the Stata code it looks like they have a number of additional cleaning steps such as those described under the overall heading ‘Additional data cleaning & duplication to ensure one birth episode per HESID’ that are currently not described in the appendix of the paper.

- Discussion, key findings & abstract – when the authors say the proportion of babies with hospital readmission after birth increased by a third to 17.7% is this compared to the proportion with hospital readmission in Q1 1998? If so, can the authors make this explicit in both the discussion and abstract or consider instead stating what I think is probably the more informative figure of 6.1% compared to the expected value based on the trend before Q4 2002.

- Page 4 of the discussion - From the data the authors have it cannot be stated with certainty that babies with longer birth admission were more likely to have their NHS number updated during the hospital stay. Also are the 35% and 25% figures quoted on page 4 of the discussion compared to the hospital readmission proportions in Q1 1998? If so, can the authors make this explicit. However, I again would consider the 5.9% and 8.6% figures (compared to expected value based on trend before Q4 2002) that the authors quote in the results to be the most relevant – and these figures actually imply a greater shift in the readmission rate occurred for births with longer not shorter birth admissions.

Other minor points:

- Abstract discussion – even if births prior to 2003 are not correctly re-linked, HES has the potential to provide national longitudinal hospitalisation birth cohort data for child research so suggest slightly reword last sentence to something like “HES has the potential to provide national longitudinal hospitalisation birth cohort data for child health research, but births prior to 2003 need correctly re-linking to follow-up records.”

- Reference 6 relates to a data linkage study conducted in Scotland so would not cite it with reference to Canada as have done in the introduction.

- Introduction – please make it explicit that HES includes all births in English NHS hospitals and presumably does the 97.4% figure relate to the proportion of all births in England rather than England and Wales?

- Methods - suggest rephrasing first sentence under study participants to something like “We developed a cohort of singleton live births between 1st January 1998 and 31st December 2015 to mother’s resident in England based on birth…”

- Methods – you state that you cleaned data on maternal age but this is not detailed in appendix 2.

- Can you make it a bit more explicit in the methods that you did not use information recorded in the mother’s delivery records in this study.

- It would helpful for completeness to include the details of the HES field/variable names you used to identify the risk factors in the appendix.

- In the methods section you state that you defined hospital admissions as a continuous period of time that a child spent under hospital care and that hospital transfers and admissions within 1 day of each other were treated as one inpatient admission which seems to contradict with what you say in appendix 3 (hospital admission defined as total time spent by a patient in one hospital, with hospital transfers classified as separate inpatient admissions) – please clarify.

- In the outcomes section of the methods where you define infant deaths suggest rephrasing slightly for clarity to: ‘Infant deaths were defined where a linked death record was found (that is, via link to ONS mortality record) or the discharge method in the hospital record was recorded as ‘died’.

- Methods – the first time you mention the implementation of NHS numbers for babies, did you mean “Q4 2002..” rather than “Q3/Q4 2002..”?

- Can you clarify that you were looking at hospital readmission in the first year of life in the methods outcomes section and appendix table 6.

- Figure 3 – would be helpful to mark the time points when the collection of identifiers used to generate HESID changed as you did for Figure 2.

- Page 2 of discussion – think you need to add an ‘of’ after ‘Further work is needed to evacuate the quality”

- On page 4 of the discussion would suggest softening the wording slight to something like “Fixing the postcode extraction error in 2013 did not appear to impact on quality of linkage, but…” Also, according to your Figure 1, fixing the postcode extraction error did not ensure that IMD was available for all births – it only correlated with an increase in the completeness of this variable to 89% in years 2014-2015 – can you amend the wording on page 4 of the discussion to reflect this.

- It would probably be clearer to use different colours rather than different shades of grey in the figures.

6. PLOS authors have the option to publish the peer review history of their article (what does this mean?). If published, this will include your full peer review and any attached files.

Reviewer #1: No

Reviewer #2: No

---

## [Author Response · Author response to Decision Letter 0]

2 Nov 2020

We would like to thank both reviewers for their comments, which helped us to revise and improve the manuscript. 

Comments to the Author - Reviewer #1:

1. This is a manuscript describing hospital episode statistics (HES) data on births in the UK and assessing whether this data can be used for following up children using routine data. While this paper provides interesting descriptive information about England’s HES statistics, the research question (s) are difficult to discern. It reads, at times, more like a report describing the constitution of a dataset, than a scientific study. The authors need to be more straightforward about the central premise of their study, clarify the research aims and place the results in a broader international context.

We thank reviewer for their comments. We have now revised the paper to more clearly reflect the aim of the paper, which is to evaluate how changes to data collection in HES and in vital statistics data over time have affected the quality of linkage between birth admissions and longitudinal follow-up records. Specific changes are listed below:

- We have clarified in the abstract and introduction that the overall aim of this study is to evaluate how changes to data collection in HES and vital statistics data over time have affected the quality of linkage between birth admissions and longitudinal follow-up records. 

- We have changed the short title to “Impact of changes to data collection on a national birth cohort from administrative health records in England”

- In the methods section, we include “Changes to collection of patient identifiers used for linkage” in a separate section to highlight that this is main exposure of interest for all outcomes. 

- We have rephrased subheadings used in the statistical analysis section of methods and results to match our aim. 

- We now explicitly refer to the three changes in data collection when describing results for infant mortality (we also highlight these changes in figure 3 and table 2) 

- In the discussion, we have revised the key findings to focus on impact of the three changes to data collection

- We have also re-organised the interpretation section of the discussion to better reflect the aim of this paper.

We also agree that the paper was lacking international context. We have now revised the introduction and discussion to provide a comparison of hospital discharge datasets available in other countries:

- In the introduction we discuss the advantages of population-level birth cohorts (1st paragraph of the introduction), we describe datasets available in England and provide a rationale for deriving a birth cohort using HES. 

- In the “interpretation” section of the discussion (paragraph 1, p22), we compare information available in HES to that collected in other countries and cite international comparative studies that include birth cohorts from HES. 

2. The introduction and title of the MS place emphasis on the methods for identifying a “national birth cohort” – the authors begin by citing the failure of the 2015 effort to establish the Life study and then refer to examples from other countries where register data are used for research. From the onset, the authors need to clarify what they mean by “national birth cohort”. There is a large difference between being able to link hospital data and studying some longer-term (mainly hospital-related) outcomes and creating a national birth cohort (as initiated in many countries, see https://lifecycle-project.eu/). When the authors claim in their discussion that they have shown the feasibility of using HES to create a national birth cohort, this is not convincing if they mean this cohort to be an alternative for cohort studies such as Life study or if comparing with the Nordic registers. Many other issues remain outstanding such as whether data can be linked to other sources, issues of consent, etc…

We thank reviewer for this useful comment. We did not intend to suggest that national birth cohorts from administrative health records can replace the traditional, nationally-representative cohorts such as the Life Study, which collected much more detailed variables and (often) biosamples for a smaller number of participants. We have now revised the introduction accordingly – we removed the mention of Life Study, and we define national birth cohorts from administrative health databases in the first paragraph of the introduction. 

3. The research questions underlying the study aims are not clear. It is stated: “In this study, we present methods for developing a national birth cohort using HES and provide Stata code for cohort derivation. We demonstrate how long-term health outcomes of children in the cohort (such as hospital admission or death) are affected by changes in the quality of recorded identifiers used for linkage of birth admissions to consecutive hospital admissions within HES and to other datasets. We suggest how linkage error can be addressed retrospectively to create hospital record trajectories for children and adolescents from 1997 onwards with accurate linkage to their birth episodes.” 

We have now clarified that the aim of this study is to evaluate how changes to data collection in national registration systems and HES have affected the quality of linkage between birth admissions and subsequent HES and mortality records, and we then list steps we took to achieve this aim. The last paragraph of the introduction (p5) now reads as follows:

“This study aims to evaluate how changes to data collection for HES and vital statistics systems over time have affected the quality of linkage between birth admissions and subsequent HES and mortality records. First, we describe methods for developing a birth cohort using HES. We then demonstrate how changes to collection of patient identifiers used for linkage of birth admissions to subsequent HES and mortality records have affected estimates of readmission and mortality rates. Finally, we suggest how linkage error can be addressed retrospectively to create hospital record trajectories in England for children and adolescents from 1997 onwards with accurate linkage to their birth episodes”

4. Concerning the first aim, presenting methods and giving code is transparent and helpful for other researchers, but does not constitute a research question. So, although instructions to identify births are provided, there is no test to show how this definition improves on others. There is no discussion about whether these methods are similar to those used in other datasets (for instance: Kuklina EV, Whiteman MK, Hillis SD, et al. An enhanced method for identifying obstetric deliveries: implications for estimating maternal morbidity. Matern Child Health J. Jul 2008;12(4):469-477). Most studies using hospital discharge data use algorithms to identify births.

We have now revised the aim of the paper, as discussed in point 3 above. The first step required to meet this aim is to develop a birth cohort using Hospital Episode Statistics. Although this is not strictly an aim of this paper, our description of how to develop a birth cohort within HES and the accompanying Stata code will be helpful for researchers interested in using HES for child health research (as pointed out by reviewer 2), which they can use and amend according to their research needs. 

In the discussion we compare our method to other proposed methods applied to HES data (first paragraph under “Strengths & limitations” heading). Comparisons with methods applied in other countries is less relevant to this work. While our algorithm includes some ICD-10 codes (which could be applied internationally), it also covers a number of data fields that are specific to HES and not applicable to other datasets (such as patient classification variable or admission method variable – coding of this variable will likely vary between hospital datasets internationally). Similarly, we note that the paper cited by the reviewer use the ICD-9 Clinical Modification system which is not used in any UK country. Further, in most countries, identification of births does not require complex algorithms, as they are indicated via linkage to birth certificates / birth records (e.g. medical birth registers in the Nordic countries). 

5. Second, the section on how changes to the quality of recorded identifiers improved linkage and quality of indicators relying on linkage isn’t particularly surprising and therefore, while it is reassuring to find that better linkage seems to improve the accuracy of some indicators (leading to higher admission rates, for instance), it’s not clear how this adds to overall knowledge about how hospital episode statistics can be used for research on children. Furthermore, there is no gold standard - in the case of hospital admissions – and there may still be substantial errors.

Since HES is collected for financial purposes rather than research, changes to data collection that are outside the control of researchers could lead to misleading conclusions (as demonstrated by the declining trend in hospital admissions before 2002 due to missed links between births and hospital admissions). There is no information on the quality of linkage from the data provider, nor information about the proportion of records with complete patient identifiers used for linkage. Therefore careful validation studies, including of any linkage between HES and other databases, need to be carried out by researchers who use these data. 

Documenting strengths and limitations of HES and methods for evaluating quality of linkage without having full information on personal identifiers used for linkage will be extremely valuable to the growing number of researchers using HES. A quick search in PubMed for mentions of “Hospital Episode Statistics” in titles and abstracts revealed that over 1000 papers have used HES since 2010 and the number of papers is growing over time. Our paper highlights that follow-up for births before 2002 is not reliable, which can help researchers to assess feasibility of their proposed studies before starting a time consuming and resource intensive application for HES data. 

We are also not clear what the reviewer means regarding no existing gold standard for measuring hospital admissions. While this may be the case in other countries, HES is considered the gold standard hospital admission dataset in England since the vast majority of acute and planned care, particularly for children, takes place within NHS hospitals and would therefore be recorded in HES (we now mention in the methods section that HES covers estimated 98–99% of all hospital activity in England). 

6. In terms of the conceptualisation of the study, the database constituted by the authors only allows a partial evaluation of the capabilities of the HES data as mothers are not linked with their babies, even though the authors state that this is possible and would improve the quality of data. The authors need to justify why they developed this study without linking these data.

This paper aimed to assess the impact of changes to data collection practices on readmission and mortality rates among infants in a HES birth cohort. While it is definitely possible to link mothers and babies within pseudonymised HES data, this paper did not aim to describe this linkage which is detailed elsewhere. We point the reader to a relevant paper which present methods for deterministic and probabilistic linkage of mothers and babies (paragraph 1 on page 21 in the “interpretation” section of discussion). Such linkage is not carried our routinely by NHS Digital, the HES data provider, which we now highlight in the paper (last sentence on p6, first paragraph on p8), and therefore some researchers might only have access to data on babies. 

7. Given the large amounts of missing data, it is not clear why the authors did not describe the characteristics of births with and without missing data – is this related to the hospital? The region?

Patterns of missing data differ between variables. As we mention, IMD score was missing due to a data extraction error for all episodes of care marked as birth. Completeness of variables in the “baby tail” such as birth weight, gestational age, maternal age varies by hospital since some hospitals submit these variables to NHS Digital and some do not (submitting these variables to NHS Digital is not mandatory ). We now mention in the discussion that rates of missing data vary between hospitals and we provide a number of references for how missing data on baby characteristics has been dealt with in other studies. For example, researchers have developed approaches to indicate hospitals with high quality of recorded data, and validate and analyse data from “sub-cohorts” of births in those hospitals with high quality of data (last paragraph on p20) 

8. The reader also wonders whether there has there been any validation of these data with medical records to assess the validity of the data?

Validation has been attempted for specific conditions, e.g. selected mental health disorders (https://www.ncbi.nlm.nih.gov/pmc/articles/PMC5868851/), but not more generally. However, diagnostic information in HES is entered by clinical coders, who are required to complete national accreditation training to ensure standardisation of recorded information between hospitals. Clinical coders translate medical notes into ICD-10 codes once patient is discharged from hospitals. Nonetheless, coding sensitivity could vary according to the quality and level of detail covered in medical notes. We have now added this information in methods under “Hospital Episode Statistics” heading (paragraph 1, p6):

9. Many countries use hospital discharge data for research; putting the HES within this broader international context would be useful. How do these results compare to those in other countries?

We agree that the paper was lacking international context. We now revised introduction to reflect our ambition to derive a data resource comparable with population-level registers available in e.g. the Nordic countries (paragraph 1, p4). We explain, that comparable data source linking birth registration, maternity and hospital admission records has only been carried out for one study to date in England, and this data resource is not updated routinely (paragraph 2, page 4). Instead, researchers are increasingly using HES to develop whole-country birth cohorts. (paragraph 1, page 5). In the discussion, we also compare the HES birth cohort data resource to those available in other countries (last paragraph of the “interpretation” section, p 22).

10. Abstract –“Numbers for Babies (NN4B) system for allocation of unique National Health Service (NHS) number at birth in Q4 2002” is not interpretable as a stand alone sentence.

Thank you for pointing this out. We have rephrased it for clarity as 

“We identified three changes to data collection, which could affect linkage of births to follow-up records: (1) the introduction of the “NHS Numbers for Babies (NN4B)”, an on-line system which enabled maternity staff to request a unique healthcare patient identifier (NHS number) immediately at birth rather than at civil registration, in Q4 2002; (2) the introduction of additional data quality checks at civil registration in Q3 2009; and (3) fixing a postcode extraction error for births by the data provider in Q2 2013.”

11. While multiples pose many problems for linkages and use of administrative and register data, the solution of eliminating them is not optimal.

We agree with the reviewer and we point this out as an important limitation of this study in the discussion (paragraph 2 on page 19). Further work is needed to evaluate the quality of linkage of birth records to hospital admissions and mortality records for multiple births, however, this was beyond the scope of this paper. Some of the challenges associated with multiple births include babies being allocated the same patient identifier, making it impossible to distinguish hospital trajectories, and challenges with recording of stillbirths. We now also refer to a recent national audit of maternity and perinatal care for women with multiple births and their babies has identified case ascertainment of 89.5% using multiple detailed datasets (Discussion page 19). 

Comments to the Author - Reviewer #2:

This study described methods used to create a national birth cohort using HES and aimed to evaluate the quality of linkage between births and follow-up records and its impact on two health outcomes in children. Overall, I think this is a well written and informative study. To help further improve the manuscript it would be good if the authors could address the following points:

1. It’s great to see that the authors have provided a link to the Stata code for derivation of the cohort. Can they just check that this is complete and consistent with all the steps they describe in the paper e.g. from a quick scan of the Stata code I could not see any code for excluding non-English residents. Similar, it would be really helpful if they can make sure that all the data cleaning steps in the Stata code are described in the appendix of the paper e.g. from the Stata code it looks like they have a number of additional cleaning steps such as those described under the overall heading ‘Additional data cleaning & duplication to ensure one birth episode per HESID’ that are currently not described in the appendix of the paper.

We thank reviewer for checking our Stata code on Github, this is greatly appreciated. We have now added extra description of our code in S2 Appendix and we have added additional do-files on GitHub for completeness. Specific changes include:

- We now refer to GitHub Repository in the appendix. We have numbered sections of the appendix and we highlight which do-file from GitHub relates to work described in each section.

- We added a brief introduction mentioning preliminary data cleaning for hospital admission and infant mortality records listed in do-files 1 and 2. 

- We have expanded section 1 of the appendix on developing the birth cohort to cover more detailed description of data cleaning to derive one birth episode per HESID, and additional information about cleaning of implausible values of birth weight, gestational age and maternal age. All steps are included in do-file 3.

- We have added additional Stata do-files describing cleaning and linking hospital admissions in the first year of life (do-file 4, described in section 2 of S2 appendix), cleaning and linking ONS mortality records (do-file 5, described in section 3 of S2 appendix) 

- We also added do-file 6 with steps taken to finalise the cohort (such as linking in additional variables derived from linked hospital admission and mortality records, deriving length of birth admission, and excluding non-English residents). These steps are described in section 4 of S2 appendix.

2. Discussion, key findings & abstract – when the authors say the proportion of babies with hospital readmission after birth increased by a third to 17.7% is this compared to the proportion with hospital readmission in Q1 1998? If so, can the authors make this explicit in both the discussion and abstract or consider instead stating what I think is probably the more informative figure of 6.1% compared to the expected value based on the trend before Q4 2002.

Thank you for highlighting this, we have now rephrased abstract and results (p14-15) to indicate that we refer to an increase by 6.1% percentage points compared to the expected value based on trend before Q4 2002. We have made additional minor changes throughout results section (p13-14) to clarify how model results should be interpreted. We have revised the “key findings“ section of the discussion, and the specific figures are no longer included. 

3. Page 4 of the discussion - From the data the authors have it cannot be stated with certainty that babies with longer birth admission were more likely to have their NHS number updated during the hospital stay. Also are the 35% and 25% figures quoted on page 4 of the discussion compared to the hospital readmission proportions in Q1 1998? If so, can the authors make this explicit. However, I again would consider the 5.9% and 8.6% figures (compared to expected value based on trend before Q4 2002) that the authors quote in the results to be the most relevant – and these figures actually imply a greater shift in the readmission rate occurred for births with longer not shorter birth admissions.

We thank reviewer for highlighting this, we agree that it cannot be stated with certainty that babies with longer birth admissions were more likely to have their NHS number updated during the hospital stay and we removed this from the discussion. Instead, in the results section we now state: “Results separated by length of birth admission were comparable suggesting that linkage error was not affected by length of stay” before stating ITSA model results (last paragraph p 13). 

Other minor points:

4. Abstract discussion – even if births prior to 2003 are not correctly re-linked, HES has the potential to provide national longitudinal hospitalisation birth cohort data for child research so suggest slightly reword last sentence to something like “HES has the potential to provide national longitudinal hospitalisation birth cohort data for child health research, but births prior to 2003 need correctly re-linking to follow-up records.”

Thank you, we agree and we have revised the discussion of the abstract to say “HES can be used to develop a national birth cohort for child health research with follow-up via linkage to hospital and mortality records for children born from 2003 onwards. Re-linking births before 2003 to their follow-up records would maximise potential benefits of this rich resource, enabling studies of outcomes in adolescents with over 20 years of follow-up.”

5. Reference 6 relates to a data linkage study conducted in Scotland so would not cite it with reference to Canada as have done in the introduction.

Thank you for spotting this error, we have now included a more relevant reference.

6. Introduction – please make it explicit that HES includes all births in English NHS hospitals and presumably does the 97.4% figure relate to the proportion of all births in England rather than England and Wales?

Thank you for this suggestion. Previously cited figure referred to England and Wales, but we have now double-checked the figure for deliveries in England only. The sentence now reads: “HES covers details of all patient care funded by the National Health Service (NHS) in England, including all births that occur in NHS hospitals (in 2016, 97.4% of all deliveries in England occurred in NHS hospitals).”

Please note, that later on we compare mortality to national statistics reported for England and Wales as mortality rates for singleton live births (rather than all live births) are not published for England only. We highlight that in the methods section and state that 95% of births in England and Wales occur in England (page 10, paragraph 1).

7. Methods - suggest rephrasing first sentence under study participants to something like “We developed a cohort of singleton live births between 1st January 1998 and 31st December 2015 to mother’s resident in England based on birth…”

Thank you, we have now rephrased this sentence to say “We developed a birth cohort of singleton live-born infants, who resided in England and were born between 1st January 1998 and 31st December 2015, based on birth admissions recorded in HES”

8. Methods – you state that you cleaned data on maternal age but this is not detailed in appendix 2.

Thank you for highlighting this omission, we have now added information about cleaning of maternal age to appendix, and we double checked that this is also included in Stata code on GitHub (see S2 Appendix section 1 and do-file 3)

9. Can you make it a bit more explicit in the methods that you did not use information recorded in the mother’s delivery records in this study.

We have added a sentence to clarify this in “study participants” section of the methods section (p8), which reads as: “We identified birth admissions using broad selection criteria based on diagnostic codes and admission details recorded in HES (such as admission type, see appendix table 3 for details). We used only information recorded in baby’s birth record, as linkage between maternal delivery and baby’s birth admissions is not routinely available”

10. It would helpful for completeness to include the details of the HES field/variable names you used to identify the risk factors in the appendix.

Thank you for this suggestion, we have now added variable names for all variables mentioned in the appendix, for example for birth characteristics we say: 

“Next, we removed implausible values of gestational age (we replaced gestational age to missing if gestational age was <22 or >45 weeks, gestat variable in HES), birth weight (replaced to missing if <200g, birweit variable in HES) and maternal age (replaced to missing if <10 or >60, matage variable in HES).”

11. In the methods section you state that you defined hospital admissions as a continuous period of time that a child spent under hospital care and that hospital transfers and admissions within 1 day of each other were treated as one inpatient admission which seems to contradict with what you say in appendix 3 (hospital admission defined as total time spent by a patient in one hospital, with hospital transfers classified as separate inpatient admissions) – please clarify.

Thank you for highlighting this inconsistency, the statement in the manuscript was correct and we have now updated information in section 2 of S2 Appendix to say:

“An admission was defined as a continuous period of time that a child spent under NHS hospital care. Hospital transfers and admissions within 1 day of each other were treated as one inpatient admission”

12. In the outcomes section of the methods where you define infant deaths suggest rephrasing slightly for clarity to: ‘Infant deaths were defined where a linked death record was found (that is, via link to ONS mortality record) or the discharge method in the hospital record was recorded as ‘died’.

Thank you, we have now rephrased this sentence as follows: “Infant deaths were indicated if a linked ONS mortality record was found or if discharge method in hospital record indicated death” (p8)

13. Methods – the first time you mention the implementation of NHS numbers for babies, did you mean “Q4 2002..” rather than “Q3/Q4 2002..”?

Thank you for highlighting this, yes, we meant Q4 2002 and we have now corrected this.

14. Can you clarify that you were looking at hospital readmission in the first year of life in the methods outcomes section and appendix table 6.

We have now clarified this in methods section and appendix table 6 (now S3 Appendix Table A).

15. Figure 3 – would be helpful to mark the time points when the collection of identifiers used to generate HESID changed as you did for Figure 2.

Thank you, we have now added lines to the figure to indicate time periods when data collection has changed, and we added a detailed caption listing these time points. 

16. Page 2 of discussion – think you need to add an ‘of’ after ‘Further work is needed to evacuate the quality”

Thank you for highlighting this, we have now corrected this sentence.

17. On page 4 of the discussion would suggest softening the wording slight to something like “Fixing the postcode extraction error in 2013 did not appear to impact on quality of linkage, but…” Also, according to your Figure 1, fixing the postcode extraction error did not ensure that IMD was available for all births – it only correlated with an increase in the completeness of this variable to 89% in years 2014-2015 – can you amend the wording on page 4 of the discussion to reflect this.

Thank you for highlighting this. We have now moved this sentence to the key findings (to better address our aim – to assess impact of changes in data collection on linkage of births to follow-up records), which now reads:

“Fixing the postcode extraction error in 2013 did not impact the quality of linkage, but it helped to improve the completeness of the IMD score for birth records in HES.”

18. It would probably be clearer to use different colours rather than different shades of grey in the figures.

Thank you, we have now added colour to the figures

---

## [Decision Letter · Decision Letter 1]

30 Nov 2020

Developing a national birth cohort for child health research using a hospital admissions database in England: the impact of changes to data collection practices

PONE-D-20-14030R1

Dear Dr. Zylbersztejn,

We’re pleased to inform you that your manuscript has been judged scientifically suitable for publication and will be formally accepted for publication once it meets all outstanding technical requirements.

Kind regards,

Umberto Simeoni

Academic Editor

PLOS ONE

Additional Editor Comments (optional):

Reviewers' comments:

Reviewer's Responses to Questions

**Comments to the Author**

1. If the authors have adequately addressed your comments raised in a previous round of review and you feel that this manuscript is now acceptable for publication, you may indicate that here to bypass the “Comments to the Author” section, enter your conflict of interest statement in the “Confidential to Editor” section, and submit your "Accept" recommendation.

Reviewer #1: (No Response)

Reviewer #2: All comments have been addressed

2. Is the manuscript technically sound, and do the data support the conclusions?

Reviewer #1: Yes

Reviewer #2: Yes

3. Has the statistical analysis been performed appropriately and rigorously? 

Reviewer #1: Yes

Reviewer #2: Yes

4. Have the authors made all data underlying the findings in their manuscript fully available?

Reviewer #1: No

Reviewer #2: No

5. Is the manuscript presented in an intelligible fashion and written in standard English?

Reviewer #1: Yes

Reviewer #2: Yes

6. Review Comments to the Author

Reviewer #1: The authors have been very responsive to the first round of review comments and the objectives and methods of the MS are now clearly stated. This will be a useful contribution to the literature.

Reviewer #2: (No Response)

7. PLOS authors have the option to publish the peer review history of their article (what does this mean?). If published, this will include your full peer review and any attached files.

Reviewer #1: No

Reviewer #2: No

---

## [Editor Report · Acceptance letter]

4 Dec 2020

PONE-D-20-14030R1 

Developing a national birth cohort for child health research using a hospital admissions database in England: the impact of changes to data collection practices 

Dear Dr. Zylbersztejn:

I'm pleased to inform you that your manuscript has been deemed suitable for publication in PLOS ONE. Congratulations! Your manuscript is now with our production department. 

Kind regards, 

on behalf of

Dr. Umberto Simeoni 

Academic Editor

PLOS ONE